



**Soil Denitrifier Community Size Changes with Land Use Change to Perennial**
**Bioenergy Cropping Systems**
*Running Head: Soil Denitrifiers associated with Perennial Grasses*
Karen A. Thompson[1], Bill Deen[2] and *Kari E. Dunfield[1]
[1]School of Environmental Sciences, University of Guelph, 50 Stone Road East, Guelph, Ontario
N1G 2W1, Canada; [2]Department of Plant Agriculture, University of Guelph, 50 Stone Road
East, Guelph, Ontario N1G 2W1, Canada.
*Correspondence to Kari E. Dunfield:
tel: (519) 824-4120 Ext.58088, *dunfield@uoguelph.ca, fax:*519-837-0756
Original Research Article



**Abstract**
Dedicated biomass crops are required for future bioenergy production. However, the effects of
large-scale land use change (LUC) from traditional annual crops, such as corn-soybean rotations
to the perennial grasses (PGs) switchgrass and miscanthus on soil microbial community
functioning is largely unknown. Specifically, ecologically significant denitrifying communities,
which regulate $N_2O$ production and consumption in soils, may respond differently to LUC due to
differences in carbon (C) and nitrogen (N) inputs between crop types and management systems.
Our objective was to quantify bacterial denitrifying gene abundances as influenced by corn-
soybean crop production compared to PG biomass production. A field trial was established in
2008 at the Elora Research Station in Ontario, Canada (n=30), with miscanthus and switchgrass
grown alongside corn-soybean rotations at different N rates (0 and 160 kg N ha$^{-1}$) and biomass
harvest dates within PG plots. Soil was collected on four dates from 2011-2012 and quantitative
PCR was used to enumerate the total bacterial community (16S rRNA), and communities of
bacterial denitrifiers by targeting nitrite reductase (*nirS*) and $N_2O$ reductase (*nosZ*) genes.
Miscanthus produced significantly larger yields and supported larger *nosZ* denitrifying
communities than corn-soybean rotations regardless of management, indicating large-scale LUC
from corn-soybean to miscanthus may be suitable in variable Ontario conditions while
potentially mitigating soil $N_2O$ emissions. Harvesting switchgrass in the spring decreased yields
in N-fertilized plots, but did not affect gene abundances. Standing miscanthus overwinter
resulted in higher 16S rRNA and *nirS* gene copies than in fall-harvested crops. However, the
size of the total (16S rRA) and denitrifying communities changed differently over time and in
response to LUC, indicating varying controls on these communities.
**Key Words: biomass, bioenergy, miscanthus, switchgrass, corn, soy**



## 1. Introduction


Future energy needs require dedicated biomass crop production for bioethanol and combustion-
based electricity generation.  Corn (*Zea mays* L.) –soybean (*Glycine max* L.) rotations currently
dominate the landscape across Ontario and the northern US Corn Belt (Gaudin et al., 2015), and
corn grain is currently the primary feedstock for bioethanol production in Canada (Jayasundara
et al., 2014).  The C4 perennial grasses (PGs) switchgrass (*Panicum virgatum* L.) and miscanthus
(*Miscanthus* spps.) have been proposed as alternate feedstock crops to corn for biomass-based
bioenergy production due to their large biomass yields, reduced nitrogen (N) and water
requirements, decreased nutrient leaching and potential for increased soil carbon (C) storage
(Blanco-Canqui and Lal, 2009; Foster et al., 2013).  Large scale production of C4 PGs in Ontario
and the northern Corn Belt would require land use change (LUC) from existing corn-soybean
rotations to PG biomass cropping systems (Deen et al., 2011; Kludze et al., 2013; Liang et al.,
2012; Sanscartier et al., 2014).
Few studies have assessed how this LUC may influence soil microbial community functioning.
In particular, soil denitrifying communities represent an ideal subset of the soil microbial
community to target to assess changes in ecosystem functioning due to agricultural management
and LUC.  Denitrifying bacteria represent approximately 5% of the total soil microbial biomass
(Braker and Conrad, 2011) and have been identified in over 60 genera (Philippot, 2006),
encompassing a wide range of phylogenetic and functional diversity.  Multiple studies have
linked changes in denitrifier communities with plant types or development stage (Bremer et al.,
2007; Hai et al., 2009; Petersen et al., 2012), N fertilization (Hallin et al., 2009; Yin et al., 2014),
organic or conventional crop management (Reeve et al., 2010), perennial vs. annual crop land
use (Bissett et al., 2011) and C and N inputs (Bastian et al., 2009).  These studies suggest that



LUC from corn-soybean rotations to PG species may influence the soil bacterial communities
which drive soil $N_2O$ production and consumption.
Denitrifier community size has been correlated with denitrification process rates (Hallin et al.,
2009; Wu et al., 2012), and denitrification potential/potential activity (Attard et al., 2011; Cuhel
et al., 2010; Enwall et al., 2010).  Potential denitrifying activity and denitrifying community size
have also been shown to be correlated in some studies (Hallin et al., 2009; Morales et al., 2010;
Szukics et al., 2010; Throback et al., 2007); suggesting community size may indicate potential
differences in soil N processes after LUC.  Particularly, the *nosZ*-bearing community may act as
a $N_2O$ sink and counter high $N_2O$ production rates (Braker and Conrad, 2011; Philippot et al.,
2011), therefore influencing $N_2O$ emissions (Cuhel et al., 2010; Morales et al., 2010; Philippot et
al., 2011).  Relative abundances of denitrifier genes have been used to assess a soil's potential to
produce (e.g. *nirS* or *nirK*) and consume (e.g. *nosZ*) $N_2O$ via denitrification, representing a
qualitative proxy of relative $N_2O$ emission potential of a soil (Butterbach-Bahl et al., 2013;
Hallin et al., 2009; Morales et al., 2010; Petersen et al., 2012; Philippot, 2002).
LUC resulting from displacement of corn-soybean rotations by PG production may alter soil
microhabitats and therefore soil microbial N-cycling due both to extensive root and rhizome
biomass and to large leaf litter inputs to soils in perennial vs. annual systems (Dohleman et al.,
2012). Within studies targeting soil microbial communities in biomass cropping systems to date
(Hedenec et al., 2014; Liang et al., 2012; Mao et al., 2013a, 2011; Orr et al., 2015), the effects of
various management practices (e.g. N fertilization and harvest) on soil microbial community
functioning have not been an area of focus.  The effect this type of LUC may have on soil
microbial communities may depend on PG management practices in these systems.



Currently, there is no consensus regarding optimal N fertilization practices for increased yields in
PG production as yield responses can be highly variable depending on environmental conditions
and crop species (Deen et al., 2011).  Depending on downstream use, miscanthus can be either
harvested in the fall pre-frost, harvested post-frost kill, or left to overwinter as standing biomass
for harvest in the spring.  Switchgrass is commonly harvested in the fall, and is often windrowed
(cut, swathed and left on soil) over winter due to producers' limitations in collecting and storing
harvested biomass in winter (Resource Efficient Agricultural Production (REAP)-Canada, 2008;
Sokhansanj et al., 2009).   Differences in N fertilizer requirements and harvest regimes may alter
C and N inputs (Attard et al., 2011) and may influence LUC impacts on soil denitrifier
community sizes.
Our objective was to compare the effects of LUC from corn-soybean to PG biomass production
on the relative abundances of total (16S rRNA gene target) and denitrifier (*nirS* and *nosZ* gene
targets) soil bacterial communities 3-4 years after PG planting.  Soil was collected on four dates
from 2011 to 2012 from a field trial established in Ontario in 2008.  The field trial is unique in
that it consists of two PG biomass crops produced in parallel with the existing common land use
of corn-soybean rotation.  It also includes multiple N fertilization rates in both annual and
perennial systems, and varied harvest practices within PG plots.  We hypothesized that soils from
PG plots would support larger total bacterial and denitrifier communities than soils from corn-
soybean plots due to increased shoot residue return and root inputs to soils in PG systems, and
that soils from PG plots with biomass harvested in the spring would support larger total bacterial
and denitrifier communities than supported by soils from PGs harvested in the fall due to
increased root inputs and leaf loss to soil over winter.



## 2. Materials and Methods

**2.1 Site Description and Experimental Design**

A field trial was established in 2008 at the University of Guelph Research Station in Elora, ON (43°38'46.73" N and 80°24'6.66" W). The field site was cultivated on May 16th and June 6th, 2008. Switchgrass (*Panicum virgatum* L., Shelter variety) was planted on June 6th, 2008. Miscanthus (*M. sinensis* x *M. sacchariflorus*, Nagara-116 variety) was planted on June 12th, 2008, and soybean (*Glycine max* L.) was planted on June 24th, 2008 and annually rotated with corn (*Zea mays* L.). Corn was planted on May 5th, 2010; soy was planted on June 3rd, 2011, and corn was re-planted on May 18th, 2012, with annual light cultivation to prepare seedbeds for planting. In 2007, prior to trial establishment, the experimental area was planted to barley (*Hordeum vulgare* L.) The soil type is a London silt loam (Gray Brown Luvisol).

The field trial was a split-split strip plot design with three replicates. The main plot factor was PG crop or annual rotation (miscanthus, switchgrass, and corn-soybean). Main treatment plots measured 6.2 m x 26.0 m. Nitrogen fertilizer (0 or 160 kg N ha$^{-1}$) was applied in strips randomly within replicates. 160 kg N ha$^{-1}$ subplots received hand-broadcast urea fertilizer (46-0-0) in May 2011 or hand-broadcast ammonium nitrate fertilizer (34-0-0) in May 2012, after soil sampling procedures described below. Main treatments were split into two harvest timings (fall or spring) within the PG fertilizer strips only. Miscanthus plots were either harvested in the late fall season after post-frost kill, or left standing to overwinter until spring harvest. Switchgrass plots were harvested in the fall, or cut and assembled into windrows in the field for biomass removal in the spring. Spring-harvest of PGs occurred before N fertilizer was applied. Harvest methods of PG yields (dry harvested biomass (tonnes) ha$^{-1}$) are described in Deen et al. (2011).



**2.2 Soil sampling and analysis**
Baseline bulk density and carbon measurements were measured for each main plot on October
23[rd], 2010.  For bulk density, two soil cores per plot were collected at 0-5 cm depth using 2.5
cm-diameter cylindrical aluminum cores.  Cores were weighed before and after drying for 24h at
105°C (Maynard & Curran, 2007).  For soil carbon analysis, ten soil cores per plot were
collected from both 0-15 cm and 15-30 cm depths using a 5 cm-diameter soil corer on a zed-
shaped transect, and then composited per treatment plot for each depth.  Total soil carbon and
inorganic carbon were analyzed with a Leco® Carbon Determinator CR-12 (Model No. 781-700,
Leco Instruments Ltd.) following the dry combustion technique (Périé and Ouimet, 2008) on
approximately 0.300 g of dried, ground and homogenized soil (Table 1).
For molecular analyses, soil was sampled on 4 dates (May 9[th], 2011; October 30[th], 2011; May
2[nd], 2012 and October 20[th], 2012).  October sampling dates occurred before fall harvest of PG
crops, while May sampling dates occurred before N fertilizer application and after spring PG
biomass removal (Fig. 1).  Ten soil cores per plot were sampled aseptically to 15 cm depth using
a 5 cm-diameter soil corer on a zed-shaped transect, composited and kept on ice until transport
back to the laboratory.  The transect shape was chosen to encompass plot heterogeneity; at a pre-
trial study date initial analysis indicated gene abundances were not significantly different
between bulk or rhizosphere soils in corn-soybean or PG plots, possibly due to the large root
biomass/leaf loss to soils in perennial plots and residual soy/corn residue cover on soil in corn-
soybean plots.  Soil samples were divided for storage at 4°C and -20°C.
Mean values of gravimetric soil moisture (g g[-1]) are shown in Figure 1.  Soil exchangeable $NO_3^-$-
N and $NH_4^+$-N were determined for each of the soil samples by KCl extraction.  Soil samples
(10.0 g) were placed into 125 mL flasks and 100 mL of 2.0M KCl was added to each flask.



Flasks were stoppered and shaken for 1h at 160 strokes per minute; solutions were allowed to
settle and were then filtered through Whatman no. 42 filter paper (Whatman plc, ME, U.S.A).
Extractable $NO_3^-$-N and $NH_4^+$-N were determined colourmetrically with segmented flow
analyses (AA3, SEAL Analytical, Wisconsin, USA) via a cadmium reduction (Technician
Instrument Corporation, 1971), and a Berthelot reaction respectively (Fig. 2).
Figure 1 illustrates the seasonal and annual variation in daily average air temperature (°C), and
daily precipitation (mm) measured at the Elora Research Station.
**2.3 Soil DNA Extraction**
Total DNA was extracted from field-moist soil sampled from each plot (3 field replicates, n=3;
total plots = 30).  DNA was extracted in duplicate (ca. 0.250g) within 48 h of sampling as per
manufacturer's protocol using the DNA PowerSoil Kit (Mobio, Carlsbad, USA).  Duplicate
extracts were then pooled, separated into aliquots, and stored at -80˚C until use in downstream
analyses.
**2.4 Quantification of total bacteria and functional genes**
Quantitative PCR (qPCR) assays were used to enumerate the total bacterial communities (16S
rRNA gene), and communities of denitrifiers by targeting nitrite reductase (*nirS*) and nitrous
oxide reductase (*nosZ*) genes, using primer pairs 338f/518r (16S, Fierer et al., 2005),
Cd3af/R3Cd (*nirS,* Throbäck et al., 2004) and 1F/1R (*nosZ,* Henry et al., 2006).  Denitrification
*nirS* and *nosZ* gene targets represent the two most important steps in the denitrification pathway
that produce gaseous by-products, and account for a large proportion of functional N genes in
some studies (Stone et al., 2015).  The first step in denitrification that produces a gaseous N
product is the reduction of nitrate ($NO_2$-) to nitric oxide (NO), catalyzed by nitrite reductases



either encoded by the cytochrome $cd_1$ (*nirS*) or copper-containing (*nirK*) genes, which are
equivalent but have not been detected within the same species (Zumft, 1997). We chose to
quantify *nirS* because ¾ of cultured denitrifiers possess the *nirS* gene (Zumft, 1997) and some
molecular reports indicate *nirS* may dominate in abundance over *nirK* in some natural
environments (Deslippe et al., 2014; Nogales et al., 2002), indicating it may be a better-suited
target for relative characterization of potential nitrite-reducing communities than *nirK*.
Additionally, *nirK* has been recently identified in autotrophic ammonia-oxidizing species
(Cantera and Stein, 2007; Casciotti and Ward, 2001), calling into question its utility in
specifically targeting denitrifying communities. The *nosZ* target codes for nitrous oxide
reductase, which catalyzes the reduction of $N_2O$ to $N_2$ in the denitrification pathway, indicating
*nosZ*-bearing communities help to complete the N cycle and determine the $N_2O:N_2$ balance.
For each gene target analyzed, duplicate replicates were run in parallel on an IQ5 thermocycler
(Bio-Rad Laboratories, Hercules, CA, USA).  qPCR reaction mixtures contained 12.5 µL of 1x
SYBR Green Supermix, each forward and reverse primer at a final concentration of 400 nM, 1
µL of DNA template and RNase/DNase-free water to a final volume of 25 µL.  The SYBR Green
Supermix contained 100 nM KCl, 40 mM Tris-HCl, 0.4 mM dNTPs, 50 units $mL^{-1}$ iTaq DNA
polymerase, 6 mM $MgCl_2$, SYBR Green 20 nM fluorescein, and stabilizer (Bio-Rad
Laboratories, Hercules, CA, USA).
Conditions for qPCR were an initiation step at 94°C for two minutes, followed by 35 cycles of
denaturing at 94°C for thirty seconds, annealing at 57°C for thirty seconds (16S) or at 55°C for
sixty seconds (*nirS*), followed by elongation at 72°C for thirty (16S) or sixty (*nirS*) seconds.  For
*nosZ,* a touchdown protocol adapted from Henry et al., (2006) was used.  Amplicon specificity
was screened by running qPCR products on an ethidium bromide-stained gel (1% agarose, 80



volts for 20 minutes) with a 100bp ladder, which resulted in clean bands for all gene targets.  The
16S rRNA primers used are degenerate and have been cited as having 89-91% matching
efficiency to all bacteria (Bergmark et al., 2012). The primers amplify one of two conserved
regions in V3 of the SSU rRNA gene, resulting in a ca. 200 bp amplicon that is within small
enough to amplify via qPCR methodology and amplifies for most bacterial taxa (Bakke et al.,

200    2011)

Known template standards were made from cloned PCR products from pure culture genomic
DNA (*Clostridium thermocellum (*16S*), Pseudomonas aeruginosa (nirS),* and *Pseudomonas*
*fluorescens (nosZ)*) and transformed into *Escherichia coli* plasmids (TOPO TA cloning kit);
plasmids were sequenced to confirm successful cloning and transformation of the target genes.
Amplicon specificity was screened by running PCR products on an ethidium bromide-stained gel
(1% agarose, 80 volts for 20 minutes) with a 100bp ladder.  PCR amplicons of cloned gene
targets were sequenced by the Laboratory Services Department at the University of Guelph using
an ABI Prism 3720 (Applied Biosystems, Foster City, CA, USA) to confirm target identity.
In all qPCR assays, all unknown samples were amplified in parallel with a triplicate serial
dilution ($10^1$-$10^8$ gene copies per reaction) of control plasmids.  PCR assays were optimized to
ensure efficiencies ranging from 93.0-106.4%, with $R^2$s ranging from 0.990-0.999 and standard
curve slopes of -3.177 to -3.408 by testing serial dilutions of DNA extracts in order to minimize
inhibition of amplification due to humic and fulvic contaminants.  Duplicate no-template
controls were run for each qPCR assay, which gave null or negligible values.  Melt curve
analysis was used to confirm amplicon specificity.  Normalization of DNA concentrations to
gram of dry soil was used to give results on a biologically significant scale; this assumes similar



DNA isolation efficiency across samples, which is only appropriate when measuring relative (vs.
absolute) quantification, as in this study.
**2.5 Statistical Analysis**
Analysis of variance was conducted in SAS 9.3 (Carlsbad, NC, USA) using a generalized linear
mixed model (PROC GLIMMIX). The Shapiro-Wilks test was used to test for normality of data;
studentized test for residuals confirmed the absence of outliers. The probability distributions of
gene abundance data sets were log normal or highly skewed and were analyzed using an
overdispersed Poisson distribution for count data (Ver Hoef and Boveng, 2007). Bulk density,
organic carbon, total carbon, nitrate and ammonium data were log transformed when required
and fitted to the normal distribution.
Within each data set, sampling time was a repeated measure; independent and interactive fixed
effects were associated with crop/crop rotation, nitrogen application rate and harvest timing
within perennial grasses, while field replicate and its associated interactions were random effects.
The residual maximum likelihood method was employed to fit the model for all data sets.
Several covariance structures were entertained before the variance components structure was
chosen based on convergence and model fitting criteria. Individual treatment means within data
sets were compared using a post-hoc Tukey's test for all pairwise comparisons. Significant
differences among and between least-square means were determined by p-values; Ho was
rejected at $p < 0.05$.
Correlation analysis was used to assess nonparametric measures of statistical dependence
between gene abundances and $H_2O$, $NO_3^--N$ and $NH_4^+-N$ measured over time (Supplementary
Table 1). Correlation analysis resulted in multiple significant correlations between variables; as



such a principal component analysis was conducted in SAS (PROC FACTOR) on 120 samples
using a VARIMAX rotation.





## 3. Results

### 3.1 Environmental and Soil Conditions



Environmental conditions varied during the periods prior to the four soil sampling dates (Figure
1). Average air temperatures over the growing seasons (May-October) were 16.9°C and 17.3°C
in 2011 and 2012 respectively (Roy et al. 2014); air temperatures in spring 2012 were warmer
than normal and resulted in earlier emergence of PG crops compared to 2011. Precipitation was
above average prior to the May 2011 sampling date (101 mm vs. 72 mm 30-year average in April
2011 and 113 mm vs. 82 mm 30-year average in May 2011) (Roy et al., 2014). In comparison,
S. Ontario received very low precipitation in April 2012 (30 mm vs. 72 mm 30-year average) and
May 2012 (28 mm vs. 82 mm 30-year average) (Roy et al., 2014). Precipitation levels were
lower in 2012 compared to 2011 from May-August (391 mm in 2011 vs. 186 mm in 2012),
however higher than normal precipitation levels occurred in October of 2011 (129 mm vs. 77
mm 30-year average) and both September (106 mm vs. 77 mm 30-year average) and October
(127 mm vs. 77 mm 30-year average) of 2012 (Roy et al., 2014). Environmental conditions prior
to soil sampling directly impact soil gravimetric content measured at time of sampling (Fig. 1),
and could also impact soil N and soil bacterial communities.
Soil physical and chemical properties were assessed in October 2010, after only two years of
contrasting management since crop establishment in 2008. The corn-soybean rotation had higher
soil bulk density than soils from both miscanthus and switchgrass plots harvested in the fall. No
differences in total or organic soil carbon were detected between the corn-soybean rotation and
the PG treatments at either the 0-15cm or 15-30cm depth (Table 1). Soil $NH_4$-N levels did not
differ significantly between the corn-soybean rotation and the PG soils, however N fertilization



significantly increased $NH_4$-N levels in soils from fall-harvested miscanthus plots ($p<0.05$) (Fig.
2a).  N fertilization also significantly increased $NO_3$-N levels in spring-harvested switchgrass
($p<0.05$) (Fig. 2b).  From May to October 2011, soil $NH_4$-N levels increased significantly and
soil $NO_3$-N levels decreased significantly in PG soils (data not shown); a similar trend was not
observed in 2012 or for soils from the corn-soybean rotation in either year.
**3.2 Biomass Yields**
Despite significant differences in precipitation between 2011 and 2012, biomass yields of
miscanthus and switchgrass did not differ between years.  In comparison, corn grain yields were
higher in 2011 vs 2012 (Table 1).  Miscanthus produced higher yields (12.7-18.3 dry tonnes ha$^{-1}$)
than either switchgrass or corn grain, regardless of N fertilization rate or harvest timing (Table
1).  When miscanthus was left standing overwinter and harvested in the spring a slight biomass
reduction was observed in 2011 (12.7-14.6 dry tonnes ha$^{-1}$); however this was not statistically
significant at $p<0.05$   When harvested in the fall and N-fertilized, switchgrass yields were not
significantly lower (10.5-11.1 dry tonnes ha-$^1$) than miscanthus yields.  Switchgrass yields from
unfertilized plots were not significantly different if harvested in the fall or spring; however,
switchgrass yields from fertilized plots were significantly higher (ca. 3-4 dry tonnes ha-$^1$) when
harvested in the fall compared to yields obtained when switchgrass was windrowed over winter.
**3.3 Bacterial Responses to Annual and Perennial Crops and their Management**
There was no statistically significant effect of N fertilization or any significant interactions
between cropping system and sampling time on any of the targeted gene abundances.  Therefore
we analyzed the impact of each biomass crop under specific harvest management on soil
bacterial gene abundances (Table 2).  Denitrifying (*nosZ*) gene copy abundances were affected





by LUC; regardless of harvest or N management, mean *nosZ* gene copies were higher in
miscanthus plots than in the corn-soybean rotation, and *nirS:nosZ* ratios were higher in the corn-
soybean soils than in miscanthus or switchgrass soils ($p<0.05$) (Table 2).  Under fall harvesting
management, biomass crop had no impact on total (16S rRNA) gene copies or *nirS* gene copies.
However, leaving miscanthus biomass standing overwinter until spring resulted in significantly
higher 16S rRNA gene copies than observed in soils from fall-harvested biomass crops and
significantly higher *nirS* gene copies than in fall-harvested switchgrass or the corn-soybean
rotation (Table 2).
**3.3 Temporal Changes in Bacterial Gene Abundances**
Sampling date had a significant impact on gene abundances for all genes quantified (Fig. 3).
Over both sampling years, 16S rRNA gene copies were significantly higher ($5.2\text{-}5.4 \times 10^9$ gene
copies $g^{-1}$ dry soil) at fall (October) sampling dates compared to the ca. $5.5\text{-}6.4 \times 10^8$ gene copies
$g^{-1}$ dry soil quantified at spring (May) sampling dates (Fig. 3).  Populations of *nirS* and *nosZ*
denitrifiers represented ca. 1.58% and 0.26% on a gene-to-gene basis *(nirS or nosZ to* 16S*)* of the
total bacterial community (data not shown), and did not follow similar trends with time of
sampling (Fig. 3). The abundance of *nirS* gene copies was significantly higher in 2012 ($4.0 \times 10^6$
$– 1.6 \times 10^7$ gene copies $g^{-1}$ dry soil) compared to 2011 ($2.5\text{-}6.3 \times 10^5$ gene copies $g^{-1}$ dry soil),
with no significant differences between May and October sampling dates within each year (Fig.
3).  The abundance of *nosZ* gene copies were approximately $1.3\text{-}3.2 \times 10^5$ gene copies $g^{-1}$ dry
soil, but increased significantly in May 2012 to approximately $3.2 \times 10^6$ gene copies $g^{-1}$ dry soil
and dropped back to previous levels by October 2012 (Fig. 3).  Higher relative proportions of
denitrifiers (*nirS* or *nosZ to* 16S*)* were observed at spring sampling dates, when total (16S) gene
abundances decreased in comparison to fall sampling dates (Fig. 3).





Two factors were selected in the principal components analysis, which accounted for 67.73%
cumulative variance.  A scree plot was examined for breaks and factors were retained when
eigenvalues ≥ 1. Soil $NH_4$-$N^+$, soil $NO_3^-$-N, *nirS* and *nosZ* loaded on factor 1, which accounted
for 43.89% variance while soil gravimetric $H_2O$ and 16S rRNA loaded on factor 2, which
accounted for 23.84% variance (Fig.4 a and b).  Differences in soil $NO_3^-$-N and $NH_4^+$-N were
strongly related to differences in *nirS* and *nosZ* gene abundances observed between May 2011
and May 2012 sampling dates (Fig. 3 and Fig. 4), while the size of the total bacterial community
(16S rRNA) was related to soil gravimetric moisture levels (Fig. 4).
**4.  Discussion**
Denitrification is an important process contributing to the production and consumption of $N_2O$ in
soils, and  mitigation of GHGs such as $N_2O$ is required to create sustainable biomass cropping
systems (Miller et al., 2008; Schlesinger, 2013).  Changes in the potential functional abilities of
the soil microbial community may reflect changes in LUC or agricultural management and
should be considered to assess the ecological impact of biomass crop production (Hedenec et al.,
2014).  Currently, few studies have assessed soil microbial community responses to PG biomass
production systems (Hedenec et al., 2014; Liang et al., 2012; Mao et al., 2013a, 2011; Orr et al.,
2015).  The highest potential to reduce GHG emissions from biomass cropping systems is to
produce crops with high yields, such as PGs (Sanscartier et al., 2014), which offset the amount of
land required for crop production (Kludze et al., 2013).  However, if PG biomass production
negatively affects soil health as indicated by changes in the potential functioning of microbial
communities, large-scale LUC from annual to perennial biomass production may not be as
sustainable as originally proposed.  As such it is necessary to identify biomass cropping systems



that not only result in large biomass yields, but also ensure agroecosystem sustainability by
maintaining or improving ecosystem services (Orr et al., 2015), such as soil N-cycling.

**4.1 Biomass Yields of Annual and Perennial Crops**

Miscanthus and switchgrass biomass yields were within the typical range of values reported
previously in Ontario  (Kludze et al., 2013; Resource Efficient Agricultural Production (REAP)-
Canada, 2008) and Europe (Christian et al., 2008; Himken et al., 1997), despite differences in
temperature and precipitation between the two study years.  Corn grain yields were within the
lower range for reported Ontario yields (Munkholm et al., 2013), potentially due to wetter (2011)
and drier (2012) field conditions than normal over the two growing seasons (Roy et al., 2014).
Deen et al. (2011) showed increases in PG biomass yields between years 2 and 3 at our site,
whereas we measured similar yields in 2011 and 2012, indicating the PGs may have reached
maximum yield potential.
Nitrogen fertilization significantly increased corn grain yields and fall-harvested switchgrass
biomass yields, however no significant increases due to N fertilization were observed in
miscanthus or spring-harvested switchgrass biomass yields.  Potential yield increases from N
fertilization in spring-harvested switchgrass may have been offset due to leaf loss over the winter
season, as increases in switchgrass yields to N fertilization have been previously observed
(Nikièma et al., 2011; Vogel et al., 2002).  Similar to the present study, European and US field
trials have also found no response of miscanthus yields to N (Lewandowski et al., 2000;
Lewandowski et al., 2003; Behnke et al., 2012; Christian et al., 2008), and PG yields were
minimally impacted by differences in growing season conditions compared to corn grain yields
(Table 1).



Despite significant differences in biomass yields between miscanthus and corn-soybean systems,
there were no significant differences in either total or organic soil carbon between any of
cropping systems assessed (Table 1).  Sampling of soil carbon occurred only two years after PG
planting; PGs are expected to be productive for 20+ years, indicating future changes in soil
carbon levels may occur.  Additionally, Ontario-based land conversion modelling scenarios have
estimated a soil carbon decrease of 2.5% upon miscanthus establishment (Sanscartier et al.,
2014), which may have negated potential increases in soil organic carbon.  However, high
miscanthus yields most likely resulted in increases in above and below-ground plant residue
return to soils (Mutegi et al., 2010; Soil Quality National, 2006);  therefore our carbon measures
may not have reflected short-term changes in labile carbon sources that had occurred.
Regardless of management or climatic conditions, miscanthus consistently produced large yields,
emphasizing its potential as a bioenergy crop suitable for production in variable Ontario
conditions.
**4.2 Bacterial Responses to Annual and Perennial Crops and their Management**
Some studies in biomass cropping systems have not observed differences in soil microbial
responses between perennial and annual crop types (Mao et al., 2011), while others have
measured significant differences in microbial abundance, diversity and community structure
between these cropping types (Liang et al., 2012; Morales et al., 2010; Watrud et al., 2013).
Currently, we observed significantly higher *nosZ* gene copies in miscanthus soils compared to
corn-soybean soils, illustrating a distinct effect of LUC from corn-soybean to miscanthus
production on soil N cycling (Table 2).



Due to the large biomass produced by miscanthus compared to corn, a large amount of plant
residues are returned to the soil; these residues provide surface cover, decrease soil bulk density,
increase water retention and regulate temperatures  (Blanco-Canqui and Lal, 2009).  Previous
work at the Elora Research Station found an inverse correlation between field-scale $N_2O$ fluxes
and *nosZ* transcript abundance in conventionally-tilled corn plots with residues returned to soils
(Németh et al., 2014), and increased *nosZ* activity after residue amendment has also been
observed in lab studies (Henderson et al., 2010).  High C:N plant residues have been negatively
correlated with cumulative $N_2O$ emissions (Huang, 2004), and may encourage complete
reduction of $N_2O$ to $N_2$ as soil available $NO_3$-N is limiting, so bacterial populations with the
ability to reduce $N_2O$ to $N_2$ are favoured (Miller et al., 2008).  Presently, the primers used for
*nosZ* gene target amplification provided good coverage of γ -Proteobacteria (Henry et al., 2006),
which are stimulated by surface-applied residues (Pascault et al, 2010).  Increased residue return
in miscanthus plots may have selected for bacterial populations harbouring enhanced catabolic
capabilities, such as $N_2O$ reduction (Pascault et al., 2010).  This implies that producing biomass
crops with large yields may indirectly alter soil N cycling and potentially mitigate soil $N_2O$
emissions due to increased residue return influencing the soil microbial community.  It is likely
that differences in environmental conditions (e.g. temperature, $H_2O$ and $O_2$ availability) and
resource quality and availability between corn-soybean and miscanthus soils related to
differences in microbial community structure (Cusack et al., 2011) and selected for different
dominant taxa that filled different ecological niches (Stone et al., 2015).
N fertilization did not affect targeted gene abundances, however studies in other cropping
systems have found that N fertilization affected the size of denitrifying communities (Hallin et
al., 2009), nitrifying communities (He et al., 2007), and proportions of *nirS/nirK* communities



(Hai et al., 2009). Elevated 16S and *nirS* gene copies were observed in soils from spring-
harvested miscanthus and windrowed switchgrass (Table 2). Increased N return via senescent
leaf loss in PG plots over winter contributes to the soil organic matter pool (Heaton et al., 2009),
and may have contributed to elevated total (16S) bacterial populations in these soils,
concomitantly increasing *nirS* abundances (Huang et al., 2011).

**4.3 Temporal Changes in Bacterial Gene Abundances**

Total soil bacterial communities (16S rRNA) followed a seasonal trend, with elevated 16S rRNA
gene copies at fall (October) compared to spring (May) sampling dates, possibly due to an
increase in the availability and diversity of resources for microbial metabolism and growth over
the growing season (Habekost et al., 2008). Denitrifying abundances changed differently than
the total bacterial community, suggesting denitrifiers were influenced by different proximal
regulators than the total bacterial community (Fig. 3 and 4). Seasonal dynamics of N-cycling
microbial communities have been previously characterized (Boyer et al., 2006; Nemeth et al.,
2014: Wolsing and Priemé, 2004; Dandie et al., 2008; Bremer et al., 2007), and are tightly
coupled with seasonal changes in labile C and N pools, temperature and soil $H_2O$ (Butterbach-
Bahl et al. 2013; Rasche et al., 2011), indicating that local edaphic drivers may often take
precedence over crop-specific drivers (Mao et al., 2013).

**5.0 Conclusions**

Miscanthus consistently produced large yields and supported larger *nosZ*-bearing communities
than the corn-soybean rotation, emphasizing its influence on soil N cycling and its potential to
mitigate soil $N_2O$ emissions while being suitable for production in variable Ontario conditions.
Additionally, miscanthus yields were not increased with N fertilization, indicating a lower N



input requirement for biomass production compared to switchgrass.  Higher 16S rRNA and *nirS*
gene copies were associated with reduced yields in spring-harvested PGs, indicating that
harvesting PGs in the spring may increase populations of denitrifiers capable of producing $N_2O$
emissions while simultaneously decreasing biomass yields.  The size of both denitrifying (*nirS*
and *nosZ)* and total bacterial (16S rRNA*)* communities changed over the sampling period,
however changes in denitrifying gene abundances did not parallel changes in the total soil
bacterial community, indicating denitrifying communities were regulated differently than the
total bacterial community.    Future work measuring $N_2O$ emissions and denitrifier activity
(mRNA) and community structure in these systems is required to link the effects of LUC on
these communities directly with $N_2O$ fluxes**.**

**Author Contributions:**

K. Thompson was the primary researcher and author on this study, conducting all field work, lab
work, and manuscript preparation.  Author B. Deen is credited for the use of his OMAFRA-
funded field plots for this research, and valuable advice on experimental design, statistical
analyses and manuscript focus.  Author K. Dunfield is credited for her invaluable mentorship on
molecular analyses and trouble-shooting on field sampling and sample preservation techniques,
manuscript focus and preparation, and data interpretation.

**Acknowledgements**

Funding for this research was awarded through grants from the Natural Science and Engineering
Research Council of Canada (NSERC), the Canada Research Chairs (CRC) Program and the
Ontario Ministry of Agriculture and Food and the Ministry of Rural Affairs (OMAF/MRA).
K.T. was also supported through an Ontario Graduate Scholarship (OGS) and an Ontario





Graduate Fellowship (OGF). Special thanks to Henk Wichers (University of Guelph) for his
expertise in managing the Alternate Renewable Fuels trial at the Elora Research Station.





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


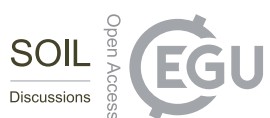

Table 1. Mean soil properties measured at the Elora Research Station.

| Cropping System/ Harvest | | N Rate | *Bulk Density | % Organic Carbon | | % Total Carbon | | Yield (dry tonnes ha⁻¹) | | |
|---|---|---|---|---|---|---|---|---|---|---|
| | | (kg ha⁻¹) | (g cm⁻³) | 0-15cm | 15-30cm | 0-15cm | 15-30cm | 2011 | 2012 | †Mean |
| Corn-soybean | Fall | 0 | 1.21 AB | 1.88 | 1.06 | 2.22 | 1.86 | 5.341 | 2.912 | E |
| Corn-soybean | Fall | 160 | 1.27 A | 1.79 | 1.47 | 2.25 | 2.11 | 9.92 | 7.882 | BC |
| Miscanthus | Fall | 0 | 1.10 B | 2.06 | 1.44 | 2.27 | 1.72 | 17.62 | 12.77 | A |
| Miscanthus | Fall | 160 | 1.10 B | 2.13 | 1.63 | 2.36 | 1.84 | 17.43 | 18.32 | A |
| Miscanthus | Spring | 0 | 1.13 AB | 2.09 | 1.53 | 2.31 | 1.69 | 12.66 | 13.38 | AB |
| Miscanthus | Spring | 160 | 1.13 AB | 2.24 | 1.42 | 2.47 | 1.89 | 14.33 | 14.56 | A |
| Switchgrass | Fall | 0 | 1.11 B | 2.12 | 1.43 | 2.33 | 1.61 | 7.648 | 6.458 | CD |
| Switchgrass | Fall | 160 | 1.09 B | 2.12 | 1.34 | 2.32 | 1.73 | 11.1 | 10.45 | AB |
| Switchgrass | Spring | 0 | 1.11 B | 2.09 | 1.23 | 2.32 | 1.55 | 6.33 | 4.146 | DE |
| Switchgrass | Spring | 160 | 1.21 AB | 1.92 | 1.33 | 2.23 | 1.7 | 6.905 | 6.441 | CD |

*Means of bulk density (n=6) followed by the same letter within one column are not significantly different according to a post-hoc Tukey's means comparison (p<0.05); carbon measurements (n=3) were not significantly different between treatments. †Mean yields (n=3) followed by the same letter are not significantly different according to a post-hoc Tukey's means comparison (p< 0.05).






Table 2. Mean gene abundance responses to crop and harvest management, averaged over nitrogen application rate and time at the Elora Research Station.

| Cropping System | Management | †Total soil bacteria (gene copy g⁻¹ soil) | †Soil denitrifying bacteria (gene copy g⁻¹ soil) | | | |
|---|---|---|---|---|---|---|
| | | 16S | nirS | nosZ | nirS:nosZ (x10⁻²) |
| Corn-Soybean | Fall Harvest | $1.35 \times 10^9$b | $1.95 \times 10^6$b | $2.63 \times 10^5$b | 7.42 |
| Miscanthus | Fall Harvest | $1.38 \times 10^9$b | $2.30 \times 10^6$ab | $4.47 \times 10^5$a | 5.15 |
| Miscanthus | Spring Harvest | $1.91 \times 10^9$a | $3.02 \times 10^6$a | $5.25 \times 10^5$a | 5.75 |
| Switchgrass | Fall Harvest | $1.41 \times 10^9$b | $2.19 \times 10^6$b | $3.55 \times 10^5$ab | 6.17 |
| Switchgrass | Spring Windrow | $1.48 \times 10^9$ab | $2.46 \times 10^6$ab | $3.98 \times 10^5$ab | 6.18 |

†Means followed by the same letter within one column are not significantly different according to post-hoc Tukey's means comparison at $p < 0.05$ (n = 24).





**Figure Captions**
**Figure 1**. Mean daily air temperature (°C) and daily precipitation (mm) at the Elora Research Station from
January 2011 to November 2012. Soil gravimetric $H_2O$ was measured on a per-sample basis and is shown as
crop means (±SE) for each sampling date (May 9th, 2011; October 30th, 2011; May 2nd, 2012 and October 20th,
2012) (n=12 in perennial grasses, n=6 in corn-soybean rotation).
**Figure 2.** Mean soil $NH_4$-N and $NO_3$-N (mg $g^{-1}$ dry soil ±SE) in annual and perennial biomass cropping
systems under varied harvest and N management at the Elora Research Station. CS = corn-soybean, SF = fall-
harvested switchgrass, SS = spring-harvested switchgrass, MF = fall-harvested miscanthus and MS = spring-
harvested miscanthus. Different letters within panels indicate significant differences according to a post-hoc
Tukey's test (p<0.05).
**Figure 3**. Mean log gene copies ($g^{-1}$ dry soil ±SE) in annual and perennial biomass cropping systems under
varied harvest management at the Elora Research Station (n=6) over time. Different letters within panels
indicate significant differences according to a post-hoc Tukey's test (p<0.05).
**Figure 4a.** Principal Component Analysis; factor 1 accounted for 43.89% variance and factor 2 accounted for
23.84% variance. **4b**. Loading plot for principal components of response variables (*nirS, nosZ* and 16S rRNA
gene copies, and soil $NO_3$-N, soil $NH_4$-N, gravimetric soil $H_2O$)..

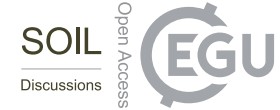


Figure 1.

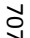

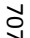

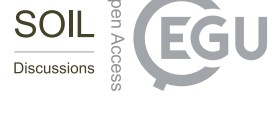

Figure 2.

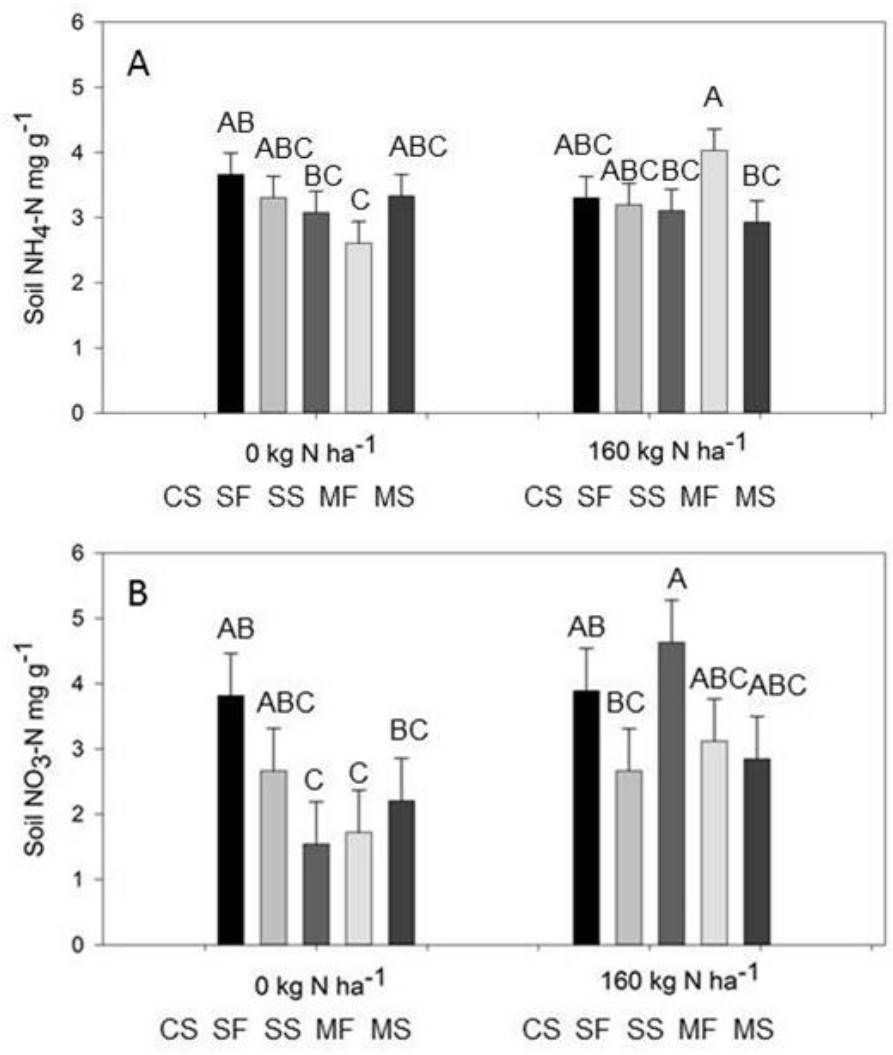




Figure 3.





Figure 4a.

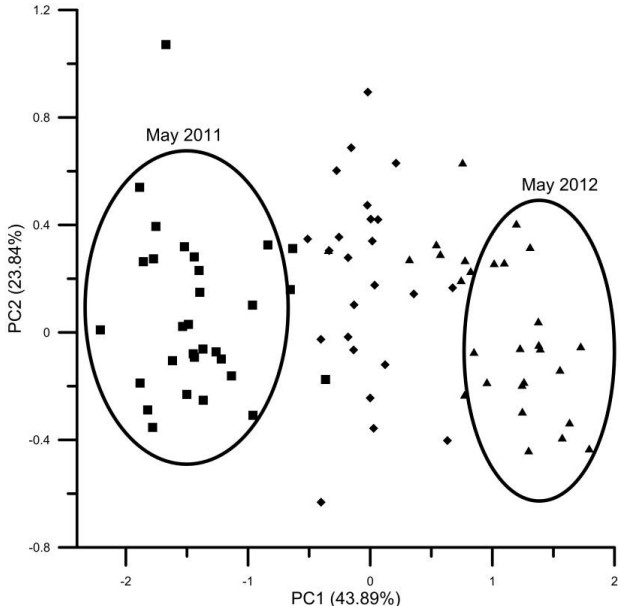


Figure 4b.

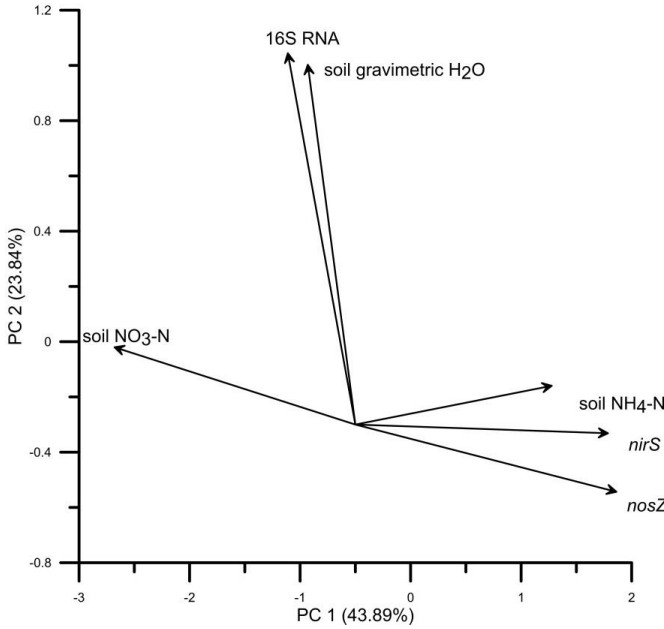
