# Peer review of "Soil Denitrifier Community Size Changes with Land Use Change to Perennial"

_SOIL, 2016_

## Short Comment (SC1) · 1 Jun 2016

This paper makes a valuable contribution to information on selection of bioenergy crops for minimum negative environmental impact, and mechanisms for how these crops affect nitrous oxide levels (which I hope we all agree should be kept to a minimum) via the soil microbiota.

---

## Short Comment (SC2) · 6 Jun 2016

This paper discuss how an ecologically important group of microbes behave in response to land use change from traditional agriculture to bio-energy crop production and highlights the suitable ways to change the land uses for bio-fuel production with potential alleviation of N2O emissions. This information is valuable for planning measures in sustaining the earth ecosystem.

---

## Short Comment (SC3) · 10 Jun 2016

General comments: This study presents valuable data to evaluate whether the growth of perennial grasses for bioenergy purposes positively or negatively affects N2O emissions compared to conventional crop rotations. Furthermore, the article also provides interesting results on biomass production with different agricultural practices and their (non-)effect on the short-term evolution of soil organic carbon. The strength of this work is the acquisition of data at the field scale within a well-designed experimental plan, and over different seasons and years. The discussion section is interesting, as authors try to contextualize all results according to a well updated literature.

Although it would have been interesting to get additional data on different parameters

(e.g. total N concentration in soil, estimation of biomass residues returning to soil for each crop), it is clear that the experimental work was already considerable here and that the authors have managed to extract the best information from their dataset. The most important results are summarized in the discussion and conclusion, both on practical aspects (recommendations to optimize miscanthus growth) and on a more process-oriented perspective (factors influencing denitrifying communities).

Specific comments: The sentence "large-scale LUC from corn-soybean to miscanthus may be suitable in variable Ontario conditions" may be removed from the abstract, as this is not the core of this paper and as this work only presents the results of one experimental site in Ontario.

Technical comments: Some abbreviations could be avoided, such as LUC and PG.

---

## Short Comment (SC4) · 21 Jun 2016

Interesting and informative findings on the use of perennial grasses as bioenergy crops. Well researched current literature. Molecular-based microbial community analysis revealed important differences in the denitrification cycling from these field trials. An important step in moving towards sustainable energy (bioenergy) with reduced GHG emissions.

Minor spelling errors:

1. page 2: "However, the size of the total (16S rRA) and denitrifying BACTERIAL communities changed differently over time"

[Figure]

2. page 8 (section 2.4): NITRITE instead of nitrate "The first step in denitrification that produces a gaseous N product is the reduction of nitrate (NO2-) to nitric oxide (NO), catalyzed by nitrite reductases"

---

## Short Comment (SC5) · 19 Jul 2016

General comments:

This article displays a well-researched and properly executed field study. The authors took their time with this study by gathering large data sets consisting of numerous of plant and soil qualities under various environmental conditions and management practices to fully understand their effects on the denitrifyer community abundance which have been shown to influence N2O emission in soil. Their objectives and consequential findings show a change to the growing of PG crops for bioenergy production to be valuable based on their positive impact on the denitrifying community responsible for N2O emission reduction compared to conventional corn-soybean rotation. High growth

yield of the PG crop regardless of management changes shows the robustness of this plant compared to conventional crops which is also a valuable finding.

Specific comments:

Line 28- Were you looking at variable Ontario conditions? I think this sentence should be changed to reflect what you looked at which was variable management practices.

Introduction- Do you have any stats on current production of PG as feedstock crops for biomass based bioenergy production?

Line 42- This seems like it should be in the introduction. Maybe the rationale for why these genes were chosen in particular could be mentioned above or significantly cut down to a sentence as part of the methods.

Line 97- Do you mean that these plots are parallel to each other in location?

Line 186-189- Explanation of the supermix components not really relevant to this article

Line 273-275- Why mention this result if "not significant"?

Line 395- Do you mean type or abundance, this is unclear?

---

## Author Comment (AC1) · 19 Jul 2016

Thank you for your review of our paper, we appreciate your comments.

---

## Author Comment (AC2) · 19 Jul 2016

Thank you for reviewing our paper and your comments.

---

## Author Comment (AC3) · 19 Jul 2016

Thank you for your assessment of our study. We assessed biomass residue return on the potential activity (mRNA transcripts) of denitrifiers over the short-term in a sub-plot RCBD study established within this experimental site, which is currently under review in another peer-reviewed journal; as such we did not include biomass/residue return measures within this paper. Thank you for your comments regarding our experimental design and data acquisition.

---

## Author Comment (AC4) · 19 Jul 2016

Thank you for reviewing our paper and pointing out some spelling errors we missed, it is very much appreciated.

---

## Author Comment (AC5) · 19 Jul 2016

Thank you for your in-depth review and valuable critique of our paper. We will respond by line to your specific comments: line 28 - climatic conditions varied by year (dry vs. wet years), however we agree that the variable management conditions should also be reflected here.

intro - we did not include stats (we are assuming you mean on biomass production rates vs. annuals?) on PG production as the focus of our paper was the soil microbiota; however there is a large base of literature on PG feedstock production in the USA, and a few studies have been conducted in Ontario (see Deen et al., 2011; Kludze et al., 2013; Liang et al., 2012; Sanscartier et al., 2014 within paper references).

line 42 - thank you for your comments, we will consider shortening this description.

line 97 - yes. This trial was unique as PG crops are often grown in large-scale (eg ha size) plots, without a parallel annual comparison at the same site/soil. Our study was a full RCBD design with smaller scale, replicated PG and annual plots grown simultaneously within the same site.

line 186 - thank you for your comments, we will consider removing the description of supermix components.

line 273 - we originally thought it was worth mentioning that miscanthus yields decreased slightly if left standing overwinter, as fertilized switchgrass yields also decreased; however your comment is valid and we will remove this result. Thank you.

line 395 - 'proportions' refers to the relative abundance of nirS: nirK denitrifiers; we will clarify this.

---

## Referee Comment (RC1) · Anonymous Referee #1 · 29 Jul 2016

The manuscript represents a good contribution to scientific progress within the scope of SOIL; it include a multidisciplinary approach and also this is good. The results are well discussed in a balanced way and conclusions are presented in a clear and cocise way; the English is appropriate. The approach and applied methods are valid even if I have some doubts about the choice of the gene used in qPCR, because of the reason I explain below.

The authors aim was to compare the effects of LUC from corn-soybean to PG biomass production on the relative abundances of total (16S rRNA gene target) and denitrifier (nirS and nosZ gene 94 targets) soil bacterial communities.

But, from literature (Case et al., 2007 Appl. Env. Microbiol. 278–288; Větrovsky

and Baldrian, 2013 PLoSONE 8(2): e57923. doi:10.1371/journal.pone.0057923) we know that the 16S rRNA gene copy numbers per genome vary from 1 up to 15 or more copies. This limits the interpretation of 16S rRNA-derived results, specially for a quantitative interpretation of the soil bacterial community. The use of a single-copy in this case would be more appropriate and could allow for a more accurate measurement of microbial community.

Thus, I suggest to the author to add some more reasons about the choice of 16SrRNA gene for bacterial quantitative purposes.

---

## Referee Comment (RC2) · Anonymous Referee #2 · 5 Aug 2016

General comments:

The manuscript from Thompson et al. is generally well structured, concise and informative. Results from their research, with higher biomass production from miscanthus but lower N2O flux, have great potential for soil science, agriculture, economics and climate change mitigations if their results can be further validated in future studies.

As the authors mentioned in Conclusions, future measurements of N2O fluxes and other relevant N cycling processes is critical in linking microbial communities to actual N2O mitigation benefits during land use change. N2O fluxes are highly variable, which raises my concern on how to interpret the information from soil denitrifier community size. Is 4 time samplings (May 9th, 2011; October 30th, 2011; May 2nd, 2012 and

October 20th, 2012) enough to represent the link between soil denitrifier community size and N2O fluxes, to differentiate seasonal changes?

Specific comments:

1, lines 61-65, confusing, need to clarify

2, lines 61-72, would it be better to add the reason why focus on N2O?

3, line 97, there are only two N fertilization rates, 0 and 160 kgN ha-1, "multiple" is not appropriate 4, line 115, add . after )

5, lines 119-120, N fertilization rates are confusing, "46-0-0" and "34-0-0" need further explanation 6, line 127, capital words in subtitles are not coherent

7, lines 155-156, strange position under 2.2 Soil sampling, suggest relocate to 2.1 Site Description

8, lines 234-235, no context for Ho

9, line 249, please explain "S. Ontario"

10, lines 243-256, authors refer to Roy et al. 2014 for result of environmental conditions instead of Fig.1. Are precipitation and temperature taken from Roy et al. 2014 ? If so, it would be better to also mention it in the Figure caption. If only soil moisture is measured, it would be better to descript soil moisture conditions instead of only mention that soil moisture "could also impact soil N and soil bacterial communities".

11, line 275, . after p<0.05

12, line 339, "years 2 and 3", please specify what 2 and 3 refer to

---

## Referee Comment (RC3) · Anonymous Referee #3 · 17 Aug 2016

The manuscript is based on the detection of nirS and nosK gene copies that could be useful to stimate the relaitve functional community variation, in the attendance with the title" Soil Denitrifier Community Size Changes with Land Use Change to Perennial 1 Bioenergy Cropping Systems.

However, as a consequence of the research approach there are some limitations to the scientific relevance of the manuscript. First, the adopted DNA soil extraction method do not permit to discriminate between relic DNA pool and the intracellular poll, without considering the PMA approach to discriminate by qPCR between relic and living cells due to contradictory results on its effciency on soil environment. Second, it is no possible to discriminate between the different nitrification/denitrification pathway and

the related microbial community. Third, It is also not possible to discriminate which of the detected species is active in the gene function without mRNA detection. Fourth, it is no possible to discriminate between the potential activity and the real activity of the nirS and nosK bacterial species. Finally result impossible to obtain extremely interesting data by coupling these data with those related to soil N2O/N2 emission.

The present contribution for the above reported consideration represent a fair study of the topic but with the suggested revision, previously sent, could be published.

---

## Author Comment (AC7) · 19 Aug 2016

The manuscript from Thompson et al. is generally well structured, concise and informative. Results from their research, with higher biomass production from miscanthus but lower N2O flux, have great potential for soil science, agriculture, economics and climate change mitigations if their results can be further validated in future studies. As the authors mentioned in Conclusions, future measurements of N2O fluxes and other relevant N cycling processes is critical in linking microbial communities to actual N2O mitigation benefits during land use change. N2O fluxes are highly variable, which raises my concern on how to interpret the information from soil denitrifier community size. Is 4 time samplings (May 9th, 2011; October 30th, 2011; May 2nd, 2012 and

[Figure]

October 20th, 2012) enough to represent the link between soil denitrifier community size and N2O fluxes, to differentiate seasonal changes?

Thank you for this comment. We believe that the timing of our sampling encompasses both seasonal changes and changes that may occur due to cropping system/management practices. Our objective was not to directly link N2O fluxes to these communities, but rather to assess whether biomass cropping systems and their management influenced the size of the denitrifier communities (ie the functional potential of these communities). Our sampling approach gave results showing significant changes in these communities based on cropping system and their management, validating our sampling choices.

Specific comments: 1, lines 61-65, confusing, need to clarify 2, lines 61-72, would it be better to add the reason why focus on N2O?

Wording of lines 61-65 have been edited to clarify (comment #1) and a sentence at the start of this paragraph has been added to provide linkage between N2O and denitrification (comment #2): "N2O is a potent greenhouse gas with a global warming potential 296x that of CO2 (IPCC 2007). However, measuring N2O directly in the field is often difficult with chamber methods in cropping systems that produce large aboveground biomass. Additionally, including multiple field treatments (eg: RCBD design) make micrometeorological methods of N2O flux impossible to obtain. Instead, relative abundances of denitrifier genes can be used to assess a soil's potential to produce (e.g. nirS or nirK) and consume (e.g. nosZ) N2O via denitrification, representing a qualitative proxy of relative N2O emission potential of a soil (Butterbach-Bahl et al., 2013; 71Hallin et al., 2009; Morales et al., 2010; Petersen et al., 2012; Philippot, 2002). Denitrifier community size has been correlated with denitrification process rates (Hallin et al., 2009; Wu et al., 2012), and denitrification potential (Attard et al., 2011; Cuhel et al., 2010; Enwall et al., 2010). Potential denitrifying activity and denitrifying community size have also been shown to be correlated with each other in some studies (Hallin et al., 2009; Morales et al., 2010; Szukics et al., 2010; Throback et al., 2007); suggesting

community size may indicate potential differences in soil N processes after LUC. Particularly, the nosZ-bearing community may act as a N2O sink and counter high N2O production rates (Braker and Conrad, 2011; Philippot et al., 2011), therefore influencing N2O emissions (Cuhel et al., 2010; Morales et al., 2010; Philippot et al., 2011)."

3, line 97, there are only two N fertilization rates, 0 and 160 kgN ha-1, "multiple" is not appropriate Within the overall field trial, there are 4 N fertilization rates (0,80, 120 and 160 kgN ha-1). We chose two (unfertilized and 160N) for assessment within our study.

4, line 115, add . after ) Thank you, done.

5, lines 119-120, N fertilization rates are confusing, "46-0-0" and "34-0-0" need further explanation Thank you, we will add in "N-P-K" to denote chemical make-up.

6, line 127, capital words in subtitles are not coherent Thank you, we will address this.

7, lines 155-156, strange position under 2.2 Soil sampling, suggest relocate to 2.1 Site Description We agree, we will move this section accordingly.

8, lines 234-235, no context for Ho Thank you, we will address this.

9, line 249, please explain "S. Ontario" Done, we will write out "Southern Ontario".

10, lines 243-256, authors refer to Roy et al. 2014 for result of environmental conditions instead of Fig.1. Are precipitation and temperature taken from Roy et al. 2014 ? If so, it would be better to also mention it in the Figure caption. If only soil moisture is measured, it would be better to descript soil moisture conditions instead of only mention that soil moisture "could also impact soil N and soil bacterial communities".

Thank you, we will edit for clarification. We use Roy et al. (2014) in text for 30 year average data, and whereas the data in figure 1 (precipitation and temperature) was collected from the Elora Research Station over the 2year study.

11, line 275, . after p<0.05 Thank you, done.

12, line 339, "years 2 and 3", please specify what 2 and 3 refer to Thank you, we will clarify this in the text – we are using 'year 2 and 3' in reference to years after LUC, so year 2 and 3 of miscanthus/switchgrass growth. We will fix this accordingly.

---

## Author Response (AR1)

September 10, 2016

MS No. soil-2016-34
To Dr. Fuensanta García-Orenes (Topical Editor, SOIL);

Please accept the following manuscript for resubmission to SOIL, entitled "**Soil Denitrifier Community Size Changes with Land Use Change to Perennial Bioenergy Cropping Systems**'', by authors Karen A. Thompson, Bill Deen and Kari E. Dunfield.

Thank you for the consideration of our original manuscript, and the constructive comments provided by the reviewers and editor. After careful consideration of the reviewer and editor comments, we have made minor revisions to the manuscript, and feel that it is greatly improved.

As specifically requested by reviewers we have added a section in the intro about our choice of the 16S rRNA gene target, made minor revisions to increase clarity, and have addressed reviewer #3's concerns in our point-by-point reply, as in the online discussion. We have also addressed concerns from the assessment of the original submission. Please refer to the Response to Reviewers for a detailed description of all of the revisions. We have also included a version of the manuscript with major changes highlighted.

Yours sincerely,

Karen Thompson and Kari Dunfield

**Responses to Editor and Reviewer Comments re: SOIL MS No.** soil-2016-34

**Topical Editor Initial Decision: Publish subject to technical corrections** (03 May 2016) by Fuensanta García-Orenes

Comments to the Author:

Please clarify how many real soil samples do you take to extract the ADN and analyze the chemical parameters, because:

the 10 samples in each treatment were mixed and two DNA samples extracted (then the two DNA extracts were mixed). This would prevent being able to see whether the variation within each treatment are larger or smaller than the variations between treatments. Also - two samples might be insufficient.

**Response: this has been clarified in the methods section; the field trial consisted of 3 replicates of 10 treatment plots (n=30).  Two subsamples were used for DNA extraction from each plot.**

Second - there is no report by the authors on the quality assurance of the primers - in particular the 16S primers. Heterogenicity (different frgment size) in the soil DNA amplification products means that quantification may be inaccurate.

**Response: This has been addressed in the methods,** "*The 16S rRNA primers used are degenerate and have been cited as having 89-91% matching efficiency to all bacteria (Bergmark et al., 2012). The primers amplify one of two conserved regions in V3 of the SSU rRNA gene, resulting in a ca. 200 bp amplicon that is within small enough to amplify via qPCR methodology and amplifies for most bacterial taxa (Bakke et al., 2011).*"

Third- The statistical analysis are not enough, please try to provide the correlations between soil properties and some PCA or cannonical analysis.

**Response: We have conducted a PCA analysis as recommended.**

**Anonymous Referee #1:**

The manuscript represents a good contribution to scientific progress within the scope of SOIL; it includes a multidisciplinary approach and also this is good. The results are well discussed in a balanced way and conclusions are presented in a clear and cocise way; the English is appropriate. The approach and applied methods are valid even if I have some doubts about the choice of the gene used in qPCR, because of the reason I explain below. The authors aim was to compare the effects of LUC from corn-soybean to PG biomass production on the relative abundances of total (16S rRNA gene target) and denitrifier (nirS and nosZ gene 94 targets) soil bacterial communities.  But, from literature (Case et al., 2007 Appl. Env. Microbiol. 278–288; Vetrovsky and Baldrian, 2013 PLoSONE 8(2): e57923. doi:10.1371/journal.pone.0057923) we know that the 16S rRNA gene copy numbers per genome vary from 1 up to 15 or more copies. This limits the interpretation of 16S rRNA-derived results, specifically for a quantitative interpretation of the soil bacterial community. The use of a single-copy in this case would be more appropriate and could allow for a more accurate measurement of microbial community. Thus, I suggest to the author to add some more reasons about the choice of 16SrRNA gene for bacterial quantitative purposes.

**Response:**

Thank you for your review of our paper.

We will adjust the language we use regarding our interpretation of our 16S rRNA results in the discussion as advised. Additionally, we will add in some text regarding our choice of 16S rRNA as follows: 16S rRNA was chosen as a molecular target for the total bacterial community size; although 16S rRNA gene copies vary from 1-15 copies per genome, its use has continued to be the 'gold standard' for microbial ecology (Case et al., 2007; Vos et al., 2012). Although an alternate target, such as rpoB, which is a single copy gene would be valuable if assessing phylogenetic diversity, there are no universal primers for it (Adékambi et al., 2009) as rpoB is not conserved enough to be of use as a universal marker and only a subset of the microbial community can be targeted (Vos et al., 2012). Therefore in order to use rpoB as a target we would have had to design a suite of different primer sets to target several orders within the same bacterial phylum, which was not feasible for this paper, and would not have measured total bacterial abundance from our diverse environmental soil samples. Taking this into account, many studies have used 16S rRNA gene copy numbers as a proxy for the total bacterial community size; and some have found that the total estimated numbers of proteobacteria species was not significantly different if using rpoB or 16S rRNA markers (Vos et al., 2012). As this study has not assessed phylogenetic relationships of the microbial communities, 16S rRNA is an appropriate target for the relative comparison of the overall bacterial community size between environmental treatments/variables.

Specifically, this text has been added to the manuscript to address this concern:

*"16S rRNA was chosen as a molecular target for the total bacterial community size; although 16S rRNA gene copies vary from 1-15 copies per genome, its use has continued to be the 'gold standard' for microbial ecology (Case et al., 2007; Vos et al., 2012). Although an alternate target, such as rpoB, which is a single copy gene would be valuable if assessing phylogenetic diversity, there are no universal primers for it (Adékambi et al., 2009) as rpoB is not conserved enough to be of use as a universal marker and only a subset of the microbial community can be targeted (Vos et al., 2012). Many studies have used 16S rRNA gene copy numbers as a proxy for the total bacterial community size, and some have found that the total estimated numbers of proteobacteria species was not significantly different if using rpoB or 16S rRNA markers (Vos et*

*al., 2012). As this study has not assessed phylogenetic relationships of the microbial communities, 16S rRNA is an appropriate target for the relative comparison of the overall bacterial community size between environmental treatments/variables."*

With these additional references:

Adékambi, T., Drancourt, M., Raoult, D., 2009. The rpoB gene as a tool for clinical microbiologists. Trends Microbiol. 17, 37–45. doi:10.1016/j.tim.2008.09.008

Case, R.J., Boucher, Y., Dahllöf, I., Holmström, C., Doolittle, W.F., Kjelleberg, S., 2007. Use of 16S rRNA and rpoB genes as molecular markers for microbial ecology studies. Appl. Environ. Microbiol. 73, 278–88. doi:10.1128/AEM.01177-06

Vos, M., Quince, C., Pijl, A.S., Hollander, M. De, Kowalchuk, G.A., 2012. A Comparison of rpoB and 16S rRNA as Markers in Pyrosequencing Studies of Bacterial Diversity 7, 1–8. doi:10.1371/journal.pone.0030600

**Anonymous Referee #2:**

The manuscript from Thompson et al. is generally well structured, concise and informative. Results from their research, with higher biomass production from miscanthus but lower N2O flux, have great potential for soil science, agriculture, economics and climate change mitigations if their results can be further validated in future studies. As the authors mentioned in Conclusions, future measurements of N2O fluxes and other relevant N cycling processes is critical in linking microbial communities to actual N2O mitigation benefits during land use change. N2O fluxes are highly variable, which raises my concern on how to interpret the information from soil denitrifier community size.

Is 4 time samplings (May 9th, 2011; October 30th, 2011; May 2nd, 2012 and C1 SOILD Interactive comment Printer-friendly version Discussion paper October 20th, 2012) enough to represent the link between soil denitrifier community size and N2O fluxes, to differentiate seasonal changes?

**Response:** Thank you for this comment. We believe that the timing of our sampling encompasses both seasonal changes and changes that may occur due to cropping system/management practices. Our objective was not to directly link $N_2O$ fluxes to these communities, but rather to assess whether biomass cropping systems and their management influenced the size of the denitrifier communities (ie the functional potential of these communities). Our sampling approach gave results showing significant changes in these communities based on cropping system and their management, validating our sampling choices.

Specific comments: 1, lines 61-65, confusing, need to clarify 2, lines 61-72, would it be better to add the reason why focus on N2O?

**Response:** Wording of lines 61-65 have been edited to clarify (comment #1) and a sentence at the start of this paragraph has been added to provide linkage between N2O and denitrification (comment #2):

"*$N_2O$ is a potent greenhouse gas with a global warming potential 296x that of CO2 (IPCC 2007). However, measuring $N_2O$ directly in the field is often difficult with chamber methods in cropping systems that produce large aboveground biomass. Additionally, including multiple field treatments (eg: RCBD design) make micrometeorological methods of $N_2O$ flux impossible to obtain. Instead, relative abundances of denitrifier genes can be used to assess a soil's potential to produce (e.g. nirS or nirK) and consume (e.g. nosZ) $N_2O$ via denitrification, representing a qualitative proxy of relative $N_2O$ emission potential of a soil (Butterbach-Bahl et al., 2013; Hallin et al., 2009; Morales et al., 2010; Petersen et al., 2012; Philippot, 2002). Denitrifier community size has been correlated with denitrification process rates (Hallin et al., 2009; Wu et al., 2012), and denitrification potential (Attard et al., 2011; Cuhel et al., 2010; Enwall et al., 2010). Potential denitrifying activity and denitrifying community size have also been shown to be correlated with each other in some studies (Hallin et al., 2009; Morales et al., 2010; Szukics et al., 2010; Throback et al., 2007); suggesting community size may indicate potential differences in soil N processes after LUC. Particularly, the nosZ-bearing community may act as a $N_2O$ sink and counter high $N_2O$ production rates (Braker and Conrad, 2011; Philippot et al., 2011), therefore influencing $N_2O$ emissions (Cuhel et al., 2010; Morales et al., 2010; Philippot et al., 2011).*"

3, line 97, there are only two N fertilization rates, 0 and 160 kgN ha-1, "multiple" is not appropriate

**Response:** Within the overall field trial, there are 4 N fertilization rates (0,80, 120 and 160 kgN ha-1). We chose two (unfertilized and 160N) for assessment within our study; however we will change the wording here.

4, line 115, add . after ) **Response:** Thank you, done.

5, lines 119-120, N fertilization rates are confusing, "46-0-0" and "34-0-0" need further explanation **Response:** Thank you, we will add in "N-P-K" to denote chemical make-up.

6, line 127, capital words in subtitles are not coherent **Response:** Thank you, we will address this.

7, lines 155-156, strange position under 2.2 Soil sampling, suggest relocate to 2.1 Site Description **Response:** We agree, we will move this section accordingly.

8, lines 234-235, no context for Ho **Response:** Thank you, we will address this.

9, line 249, please explain "S. Ontario" **Response:** Done, we will write out "Southern Ontario".

10, lines 243-256, authors refer to Roy et al. 2014 for result of environmental conditions instead of Fig.1. Are precipitation and temperature taken from Roy et al. 2014 ? If so, it would be better to also mention it in the Figure caption. If only soil moisture is measured, it would be better to descript soil moisture conditions instead of only mention that soil moisture "could also impact soil N and soil bacterial communities".

**Response:** Thank you, we will edit for clarification. We use Roy et al. (2014) in text for 30 year average data, and whereas the data in figure 1 (precipitation and temperature) was collected from the Elora Research Station over the 2year study.

11**,** line 275, . after p **Response:** Thank you, we will check this.

12, line 339, "years 2 and 3", please specify what 2 and 3 refer to **Response:** We have adjusted the language to indicated 2 and 3 years after planting.

**Anonymous Referee #3:**

Thank you for your review of our work. Please find responses to your comments below.

First, the adopted DNA soil extraction method do not permit to discriminate between relic DNA pool and the intracellular poll, without considering the PMA approach to discriminate by qPCR between relic and living cells due to contradictory results on its efficiency on soil environment.

**Response**: Although this is true, at the time this research was conducted (2011-2012), there were no published PMA protocols for environmental matrices, such as soil. Additionally, although some studies have shown an impact of relic DNA on diversity meaC1 SOILD Interactive comment Printer-friendly version Discussion paper sures (Carini et al., 2016), others have shown that despite PMA decreasing extracted DNA yields, these decreases did not have a subsequent impact on fingerprinting measures, such as DGGE (Wagner et al., 2015). In this article, we aren't comparing taxonomic diversity etc. but making comparisons of gene abundances (functional potential) within one soil type between crop treatments. Therefore, the comparisons of gene abundances are still relevant. Finally, although the use of PMA in environmental matrices is still being improved upon, the efficiency of PMA on different taxa is unknown, and PMA permeability into cells might vary across taxa, indicating that we should interpret PMA-treated data with caution. There has also been some evidence (Taylor et al., 2014) that depending on the environmental matrix assessed and extraction method, at higher concentrations PMA may bind to DNA in viable cells, leaving only dormant state microbes, and therefore not be effective in differentiating viable and non-viable cells.

Paul Carini, Patrick J Marsden, Jonathan W Leff, Emily E Morgan, Michael S Strickland,Noah Fierer. Relic DNA is abundant in soil and obscures estimates of soil microbial diversitybioRxiv 043372; doi: http://dx.doi.org/10.1101/043372

Taylor MJ, Bentham RH, Ross KE. Limitations of Using Propidium Monoazide with qPCR to Discriminate between Live and Dead Legionella in Biofilm Samples. Microbiology Insights. 2014;7:15-24. doi:10.4137/MBI.S17723.

Wagner AO, Praeg N, Reitschuler C, Illmer P. Effect of DNA extraction procedure, repeated extraction and ethidium monoazide (EMA)/propidium monoazide (PMA) treatment on overall DNA yield and impact on microbial fingerprints for bacteria, fungi and archaea in a reference soil. Applied soil ecology: a section of Agriculture, Ecosystems & Environment. 2015;93:56-64. doi:10.1016/j.apsoil.2015.04.005.

Second, it is no possible to discriminate between the different nitrification/denitrification pathway and the related microbial community.

**Response:** I think you are inferring that we cannot connect functional potential (gene abundances) with community composition/identification, or process rates. This was not our intent, and we have not attempted to directly link the denitrification pathway with gene abundance data, but have instead assessed the sustainability of these cropping systems based on functional gene abundances involved in the denitrification pathway.

Third, It is also not possible to discriminate which of the detected species is active in the gene function without mRNA detection.

**Response:** We agree that it is not possible to assess potential activity with DNA-based methods. However, mRNA has a halflife of minutes and was thought to be inappropriate for assessment of denitrifier communities due to the timing of sampling in our study (which was over 2 years). It is more plausible to assess the potential functionality of the soil microbial community to cropping systems when sampling over the longterm than attempting to link differences in mRNA with edaphic factors or agricultural treatments as mRNA may be upregulated in response to short-term (in the order of minutes, hours) changes in soil and climatic properties, while DNA analyses may allow better differentiation of changes due to cropping systems.

Fourth, it is no possible to discriminate between the potential activity and the real activity of the nirS and nosK bacterial species.

**Response:** We have not attempted to, or claimed to, assess potential activity (mRNA) or real activity (enzymes) of denitrifiers in this study.

Finally result impossible to obtain extremely interesting data by coupling these data with those related to soil N2O/N2 emission.

**Response:** Although we would have preferred to measure N2O emissions, the fieldset up did not permit this. First, the plots were too small and numerous (N=36) to establish eddy covariance/flux towers. Secondly, due to the large root biomass, above ground biomass and overall ground coverage of miscanthus and switchgrass plants, after consulting with a micrometeorologist, we were advised that it would be impossible to install chambers within our plots without highly disturbing the area, and therefore obtaining biased results. Our focus was in assessing the sustainability of the cropping systems by comparing the functional potential of the soils to produce or consume N2O by quantifying denitrifier gene targets.

[revised manuscript text omitted]